# OpenReview forum: "PlotCraft: Pushing the Limits of LLMs for Complex and Interactive Data Visualization"
_ICLR.cc/2026/Conference — ICLR 2026 Conference Withdrawn Submission_

### Official Review · Reviewer_cAYw · 2025-10-28

**Soundness:** 3
**Presentation:** 4
**Contribution:** 3
**Rating:** 6
**Confidence:** 3

**Summary:**

The paper introduces PlotCraft, a benchmark of 982 text-to-visualisation tasks spanning 7 high-level intents and 48 chart types, explicitly targeting the ability of LLMs to generate complex plotting with multiple subplots and multi-turn refinement from structured raw data. It evaluates 24 LLMs (23 external and their fine tuned model PlotCraftor) and finds substantial gaps on medium/hard visualisation tasks. To address this, the authors create SynthVis-30K, a large scale dataset of multi-modal task–code pairs created via a multi-agent pipeline and then create PlotCraftor by fine tuning (SFT) Qwen3-Coder-30B-A3B on SynthVis-30k. PlotCraftor reaches top open-weight performance and approaches strong proprietary models across PlotCraft, VisEval, and PandasPlotBench, with particularly large gains on hard tasks. The benchmark, dataset, and code are open sourced.

**Strengths:**

- **Clear problem and gap:** Complex, multi-subplot, multi-turn visualisation is under-evaluated; PlotCraft directly targets this. It also provides quite an extensive series of test, well varied.
- **Well-specified evaluation:** Two-stage pipeline (valid execution + automated visual judge) with decomposed compliance/quality metrics and reported human agreement (Cohen’s κ).
- **Breadth of baselines & analysis:** 23–24 models across difficulty tiers; insightful scaling analysis showing hard-task are not really solved with models under ~100B params.
- **Substantive details:** SynthVis-30K via multi-agent pipeline and PlotCraftor training details; reproducibility and release plan.
- High quality figures (for instance 1 and 3 are really clear and qualitative)

**Weaknesses:**

- **Judge dependence:** Primary scores come from Gemini-2.5-Pro; although correlated with humans, single-judge bias and failure on subtle visual errors remain concerns. Multiple models could be used concurrently for a more robust way of obtaining the score
- **Licensing/annotator clarity:** Ethics statement says no external workers; elsewhere, “human annotators” crafted refinement prompts—clarify who they were and dataset licence variability.
- **Scope limitations:** Focus on matplotlib/Python; portability to other viz ecosystems is not evaluated. (Authors note this implicitly as future work.)
- **Automated synthesis cost/filters:** Multi-agent generation is costly; strict termination criteria may bias examples toward what the judge favours.
- Results lack variance estimates (no std/CIs across seeds).

**Questions:**

- **Judge robustness:** How do results/rankings change if Claude/GPT-4o serve as judges, or using 2 LLMs + the human factors? Any statistically significant rank flips?
- **Human evaluation scale:** Beyond the 500-chart subset, could you provide CIs and inter-rater stats per sub-metric and per difficulty tier?
- **Licensing & annotators:** Were the “human annotators” solely the author team? Any variability in Kaggle dataset licences that restrict redistribution of derived images?
- **Generalisation:** Do findings carry over to Vega-Lite/Altair, Plotly, or ggplot2 targets? Any preliminary cross-backend experiments? Or has it not yet been tested by the authors ?
- Please add **mean±std over seeds** for main metrics (by difficulty + turn setting). If the judge is stochastic (which i would assume as you use Gemini-2.5-Pro), include **std over multiple seeds**
- Small typo to correct:
    - Title Table 3 “benchamrks”

---

> ### Author Response · Authors · 2025-11-21
>
> We sincerely thank Reviewer cAYw for recognizing the quality of our work and for providing constructive feedback. We address your concerns as follows:
>
> ### 1. Robustness, Judge Dependence, and Variance (W1, Q1, Q5)
>
> We share your concern regarding judge robustness and the lack of explicit variance reporting. Our internal research focused extensively on the reliability of automated evaluation:
>
> * **Judge Model Comparison:** We discuss the alignment of various LLM judges with human ratings in **Section 5.3** and **Appendix H**. Specifically, **Table 4, Table 8, and Table 9** compare human correlation across Gemini-2.5-Pro, Claude-4-Sonnet, and ChatGPT-4o.
> * **Judge Capability:** Our findings consistently showed that **Gemini-2.5-Pro** was the only model capable of effectively judging the complexity and nuance of these visualization tasks. Due to the inherent difficulty of the benchmark, using Claude or GPT as scoring models led to non-discriminatory results, making rankings derived from them statistically unreliable.
> * **Ensuring Robustness (Q5):** To address the lack of variance estimates, we confirm that all scores reported in **Table 2** are the average of 5 separate evaluation runs. Following your suggestion, we have added **Table 10** to the Appendix, which explicitly presents the **mean $\pm$ std** for the key metrics based on these 5 runs.
>
> ### 2. Licensing and Annotator Clarity (W2, Q3)
>
> Thank you for seeking clarification on our team and data usage.
>
> * **Annotator Identity:** All human annotators mentioned in the paper, particularly those who crafted refinement prompts, were exclusively members of the **author team**. No external workers were employed.
> * **Dataset Licensing and Scope:**
>     * **PlotCraft Benchmark (Raw Source Data):** We sourced open-source datasets from Kaggle. The use of this raw data in the benchmark is for scientific research purposes only, and its utilization is subject to the original licenses (e.g., CC-NC, CC-BY).
>     * **SynthVis-30K (Synthesized Dataset):** The large-scale synthesized dataset, SynthVis-30K, is released under CC BY 4.0 or Apache 2.0 compliant licenses to maximize its usability for model training.
> * **Data Availability:** We do not redistribute the raw source data in our public code repository. Benchmark users are directed to download the original data themselves via the provided instructions.
>
> We have emphasized these critical details in the revised version and the accompanying data documentation.

---

> ### Author Response · Authors · 2025-11-21
>
> ### 3. Human Evaluation Scale and Reliability (Q2)
>
> We thank the Reviewer for raising the important question about the robustness of our human evaluation. To confirm the high quality and reliability of our human scores, we provide the **Fleiss' Kappa ($\kappa$)** inter-rater agreement across three human evaluators, and the **95% Confidence Intervals (CIs)** for the results derived from the majority vote.
>
> #### High Inter-Rater Agreement (Fleiss' $\kappa$)
>
> The overall average $\kappa$ is **0.8884**, which is classified as "Almost Perfect" agreement. This shows strong consensus among human raters, confirming the **high reliability and validity** of our human evaluation process across all sub-metrics and difficulty levels. Agreement remains robust even for the HARD difficulty tasks.
>
> | Metrics | Overall $\kappa$ | EASY $\kappa$ | MEDIUM $\kappa$ | HARD $\kappa$ |
> | :--- | :--- | :--- | :--- | :--- |
> | **Compliance (0/1)** | **0.9237** (Avg) | 0.9184 (Avg) | 0.9396 (Avg) | 0.7852 (Avg) |
> | Layout | 1.0000 | 1.0000 | 1.0000 | 1.0000 |
> | Type | 0.9093 | 0.8504 | 0.9507 | 0.7158 |
> | Visual | 0.8880 | 0.9061 | 0.9155 | 0.7358 |
> | Task | 0.8974 | 0.9170 | 0.8921 | 0.6891 |
> | **Quality (0/1/2)** | **0.8602** (Avg) | 0.9093 (Avg) | 0.8738 (Avg) | 0.7697 (Avg) |
> | Clarity | 0.8846 | 0.9094 | 0.8796 | 0.8331 |
> | Layout | 0.8608 | 0.9182 | 0.7814 | 0.8504 |
> | Color | 0.8764 | 0.9487 | 0.9386 | 0.7343 |
> | Text | 0.8333 | 0.8859 | 0.8830 | 0.6740 |
> | Format | 0.8459 | 0.8841 | 0.8864 | 0.7569 |
> | **Overall Average $\kappa$** | **0.8884** | 0.9133 | 0.9030 | 0.7766 |
>
> #### Confidence Intervals (95% CI) for Human Scores
>
> The 95% CIs confirm the precision of the reported mean scores. These narrow intervals show that our sampled human evaluation results are highly representative of the true performance distribution, further demonstrating the **strong validity and confidence** in our human evaluation metrics.
>
> | Difficulty | Metric | Mean Score | 95% Confidence Interval (CI) |
> | :--- | :--- | :--- | :--- |
> | **EASY** （163）| Compliance (0/1) Avg | 0.7459 | N/A |
> | | Layout | 0.8834 | [0.8344, 0.9284] |
> | | Type | 0.8446 | [0.7873, 0.8978] |
> | | Visual | 0.6871 | [0.6155, 0.7546] |
> | | Task | 0.5685 | [0.4928, 0.6421] |
> | | Quality (0/1/2) Avg | 1.1953 | N/A |
> | | Clarity | 1.2311 | [1.0982, 1.3599] |
> | | Layout | 1.3395 | [1.2188, 1.4520] |
> | | Color | 1.3047 | [1.1943, 1.4131] |
> | | Text | 1.1350 | [0.9938, 1.2740] |
> | | Format | 0.9673 | [0.8344, 1.0961] |
> | **MEDIUM** （167）| Compliance (0/1) Avg | 0.4876 | N/A |
> | | Layout | 0.6946 | [0.6248, 0.7645] |
> | | Type | 0.5130 | [0.4371, 0.5888] |
> | | Visual | 0.4072 | [0.3333, 0.4790] |
> | | Task | 0.3353 | [0.2675, 0.4052] |
> | | Quality (0/1/2) Avg | 0.8203 | N/A |
> | | Clarity | 0.7345 | [0.6128, 0.8583] |
> | | Layout | 0.9681 | [0.8443, 1.0918] |
> | | Color | 0.9681 | [0.8503, 1.0858] |
> | | Text | 0.5788 | [0.4671, 0.7006] |
> | | Format | 0.8583 | [0.7345, 0.9820] |
> | **HARD** （170）| Compliance (0/1) Avg | 0.2118 | N/A |
> | | Layout | 0.6078 | [0.5353, 0.6804] |
> | | Type | 0.0725 | [0.0373, 0.1137] |
> | | Visual | 0.1118 | [0.0686, 0.1589] |
> | | Task | 0.0529 | [0.0235, 0.0863] |
> | | Quality (0/1/2) Avg | 0.5592 | N/A |
> | | Clarity | 0.4294 | [0.3412, 0.5216] |
> | | Layout | 0.6176 | [0.5294, 0.7059] |
> | | Color | 0.7216 | [0.6196, 0.8235] |
> | | Text | 0.4000 | [0.3157, 0.4882] |
> | | Format | 0.6275 | [0.5353, 0.7216] |
>
>
> ### 4. Response to Weakness: Scope Limitations (W3, Q4 )
>
> We agree with the Reviewer that extending our framework to other visualization ecosystems is an important next step.
>
> Our current focus is on Python's imperative libraries ($\text{matplotlib}/\text{seaborn}$). We note that extending PlotCraft and PlotCraftor to declarative languages ($\text{Vega-Lite}/\text{Altair}$) or different backends ($\text{R}/\text{ggplot2}$) is a key direction for future work.
>
> Critically, our methodology focuses on separating the core components:Task $\rightarrow$ Data $\rightarrow$ Chart Intent. This separation is inherently independent of any specific programming language. Thus, the design of the PlotCraft benchmark provides a solid foundation for future cross-backend benchmarking and model development.
>
> ### 5. Response to Weakness: Automated Synthesis Cost/Filters (W4)
>
> We recognize the concern regarding potential synthesis bias.
>
> To confirm the generalization ability of our model, PlotCraftor, we evaluated its performance on two external benchmarks: VisEval and PandasPlotBench. As shown in **Table 3**, PlotCraftor achieves strong results on these unseen tasks. This validates that the synthesized data trains a model with robust generalization, moving beyond the specific outputs of the synthesis pipeline.
>
> ### 6. Minor Corrections
> Typo: We have corrected "benchamrks" in the title of Table 3. Thank you for the detailed reading.

---

### Official Review · Reviewer_gbQN · 2025-10-30

**Soundness:** 3
**Presentation:** 3
**Contribution:** 3
**Rating:** 6
**Confidence:** 4

**Summary:**

This paper addresses the limitations of Large Language Models (LLMs) in generating complex data visualizations. The authors introduce three key contributions: 1) PlotCraft, a new benchmark with ~1k challenging tasks to evaluate both single-turn generation and multi-turn refinement. 2) SynthVis-30K, a large-scale, high-quality dataset of visualization code synthesized via a multi-agent framework. 3) PlotCraftor, a fine-tuned 30B parameter model that achieves performance comparable to leading proprietary models like Claude-4-Sonnet on complex visualization tasks. The work demonstrates that while current LLMs struggle with complexity, targeted fine-tuning on high-quality, domain-specific data can significantly bridge this performance gap.

**Strengths:**

- **Timely Benchmark** (PlotCraft): The paper introduces a much-needed benchmark that moves beyond simple text-to-chart tasks (e.g. VisEval, NVbench). I think this benchmark is a timely step towards better alignment of llm’s coding capability and people’s practical needs for visualization generation.
- **Strong Empirical Results and Model** (PlotCraftor): The comprehensive evaluation across 24 models provides clear evidence of the current state-of-the-art and its limitations. The proposed PlotCraftor model demonstrates impressive results, showing that a smaller, specialized model can achieve performance on par with much larger proprietary systems.

**Weaknesses:**

- **Reliance on Automated Evaluation**: While the authors validate a MLLM-based judge (Gemini-2.5-Pro) against human scores, current MLLM can still miss important visual flaws / quality aspects that can be easy for human to inspect. Moreover, relying on a certain version of proprietary MLLM (Gemini-2.5-Pro) for benchmark could lead to potential bias and stability risks. These limitation should be discussed more prominently. Some relevant works that are recommend to review and discuss:
[1] VisJudge-Bench: Aesthetics and Quality Assessment of Visualizations
[2] VIS-Shepherd: Constructing Critic for LLM-based Data Visualization Generation
[3] Closing the Feedback Loop in Text2Vis: Refining Visualization with Vision-Language Models

- **Reproducibility and Cost of Data Synthesis**: The data synthesis pipeline appears to rely on powerful, proprietary models (e.g., Claude), which may create a high cost and reproducibility barrier for other researchers. A discussion on the associated costs and the feasibility of using open-source alternatives would strengthen the paper.

**Questions:**

- Could you provide a few concrete examples where the automated visual judge's score differed significantly from human evaluators?

- Can you comment on the approximate cost (e.g., API calls/cost per sample) of your SynthVis-30K data generation pipeline? Have you explored the feasibility of replacing the proprietary models in your pipeline with open-source ones? Which model does you use for the Planner, Coder and Debugger? Could you list them explicitly in the manuscript for better transparency and reproducibility?

- Could you elaborate on the specific types of reasoning you believe are crucial for "Hard" tasks and are lacking in smaller models?

---

> ### Author Response · Authors · 2025-11-21
>
> We sincerely thank Reviewer gbQN for the thorough review and constructive feedback. We address your concerns below:
>
> ### **1. Reliance on Automated Evaluation** (W1)
>
> We fully agree that relying on a model judge introduces potential bias and risk. We have added a explicit discussion of this limitation to **Section 3.4** (Evaluation Setup) in the revised manuscript, integrating relevant insights from the papers you recommended.
>
> Regarding the choice of judge, our experimental findings align with observations in related work (e.g., VisJudge-Bench):
>
> * **Judge Model Comparison:** We conducted a careful analysis of different MLLM judges against human ratings, detailed in **Section 5.3** and **Appendix H**. Specifically, **Table 4, Table 8, and Table 9** compare the human correlation achieved by Gemini-2.5-Pro, Claude-4-Sonnet, and ChatGPT-4o.
> * **Judge Capability:** Our results consistently demonstrated that **Gemini-2.5-Pro** was the most effective judge, showing the strongest correlation with human scores and being the only model capable of distinguishing performance differences on our complex benchmark. Using Claude or GPT as scoring models led to non-discriminatory results, as they were more prone to assigning high scores to visualizations containing errors.
>
> ### 2. Automated Judge vs. Human Discrepancy Example (Q1)
>
> The MLLM visual judge excels at understanding the **macro-level semantics** of the visualization (e.g., correct chart type, data mapping) but shows limitations in judging **micro-level visual details** and aesthetics.
>
> We include a representative example of MLLM judge failure in the **Appendix, Figure 34**, where the judge assigned a significantly lower score than human evaluators.
>
> In this specific case:
>
> * **Human Evaluator Score:** Awarded 6 points on the Quality metrics (**Clarity: 2**, Layout: 1, Color: 1, Text: 1, Format: 1).
> * **Gemini-2.5-Pro Judge Score:** Awarded 4 points on the Quality metrics (**Clarity: 0**, Layout: 1, Color: 1, Text: 1, Format: 1).
>
> The reason for the MLLM visual judge's low score was:
> > "There is significant overlap in the x-axis tick labels of the top-left subplot (0,0), rendering them completely unreadable."
>
> **Correction:** We note that the x-axis labels were correctly arranged and no overlap occurred. This highlights a key failure mode where the MLLM judge incorrectly identifies a non-existent critical error, leading to an unfair low score.
>
> ### **3. Reproducibility and Cost of Data Synthesis** (W2,Q2)
>
> We appreciate your concerns regarding the cost and reproducibility of our data synthesis pipeline for SynthVis-30K. We provide the following details to clarify these aspects:
>
> * **Cost and Token Usage:** The synthesis pipeline required an average of 3.3 loops (median 3) per sample. The average token consumption per sample was approximately 10,000 tokens. The token usage ratio between the code generation agents and the judge agent was roughly 2:1.
> * **Model Configuration:**
>     * Planner, Coder, and Debugger Agents: Claude-4-Sonnet.
>     * Visual Judge: Gemini-2.5-Pro.
> * Open-Source Feasibility: We tested replacing all non-judge agents with the open-source model Qwen3-Coder. The model trained on this equivalent-sized dataset achieved an AVG score of 5.85 on PlotCraft, which remains superior to all other open-source baselines. This confirms the robustness of our synthesis approach.
> * **Judge Replacement:** At the time of this research, a suitable high-quality open-source replacement for the Visual Judge was not available. Gemini-2.5-Flash is noted as a potential cost-effective, closed-source alternative.
>
> ### **4. Reasoning Crucial for "Hard" Tasks in Small Models** (Q3)
> We observe that small models struggle significantly with "Hard" tasks, exhibiting Pass Rates consistently below 20% (except PlotCraftor). This performance gap stems from deficiencies in two specific reasoning capabilities:
>
> 1.  **Complex Layout Planning:** Small models fail to pre-calculate necessary intermediate data (e.g., global axis limits, subplot spacing) before generating code. This leads to **failed layouts**, characterized by overlapping text, misaligned subplots, or cut-off legends.
> 2.  **Handling Complex Chart Types:** They lack the reasoning to transform raw data into the specific structures required for **complex chart types** (e.g., Sankey diagrams, financial Candlesticks), often erroneously applying simple plotting logic to multi-dimensional data.

---

### Official Review · Reviewer_BuCn · 2025-10-31

**Soundness:** 3
**Presentation:** 3
**Contribution:** 3
**Rating:** 6
**Confidence:** 4

**Summary:**

This paper presents PlotCraft, a benchmark for evaluating LLM's capability to generate charting code from detailed instructions. Comparing to prior benchmarks, PlotCraft includes harder data visualization tasks that involve subplots and layers that improves upon previous benchmarks. The paper also presents a training synthetic dataset SynthVis-30k to support visualization tasks similar to PlotCraft (based on a disjoint set of Kaggle data). The evaluation shows that the model trained on this dataset perform as good as out of the box closed-source models. The benchmark also shows interesting observations on limitations of simple benchmarks and reasoning/non-reasoning model performance.

**Strengths:**

- new dataset capturing a much richer set of visualizations comparing to previous benchmarks, better for measuring model's data visualization performance.
- the synthetic benchmark provides source for finetuning models towards the goal.

**Weaknesses:**

- The task in this dataset is quite detailed and verbose, which may not match how such models would be used in practice. But on the other hand, I don't think this is a big issue since the paper focuses on model capability measurement. But this worth emphasis, since many practical visualization tasks are quite more open-ended from a high-level question.
- From examples in appendix, some charts with layout issues could be the result of the data and instruction characteristics (i.e., if the model follow instructions exactly, the chart may already be over-crowded). There could be some discussion about whether the task + data combinations may just lead to non-optimal visualizations.
- The LLM as judge evaluation, while standard, worth some deeper investigation. For example, given reasonable consistency between human and LLM judge agreement on different aspects of evaluation, whether the assemble into the final score for evaluation.

**Questions:**

- It would be ideal for authors to discuss visualization creation from detailed instruction vs abstract goal (which could lead to multiple feasible charts, not just one), and clarify the paper falls into the first category in revision.
- better elaborate LLM-as a judge quality.

---

> ### Author Response · Authors · 2025-11-21
>
> We sincerely thank Reviewer BuCn for the detailed review and constructive feedback, which have helped us improve the clarity of our paper.
>
> ### 1. Response to W1 & Q1: Detailed Instructions vs. Abstract Goals
>
> We fully agree with your observation that generating visualization code from abstract goals is a significant real-world demand. We acknowledge this distinction and offer the following clarification:
>
> * **Reason for using detailed instructions:** We conducted preliminary experiments with abstract instructions. We found that abstract prompts often lead models to generate simple charts that lack specific details, making it difficult to objectively evaluate the quality of the output.
> * **Focus on Capability:** Performance under abstract instructions tends to reflect a model's *generation preference* (what it chooses to do) rather than its *capability limit* (what it is able to do). Since PlotCraft aims to benchmark the upper limits of complex code generation, detailed instructions are necessary to ensure a rigorous stress test.
>
> **Revision:**
> Following your suggestion, we have added a discussion regarding the instruction level and the distinction between these two paradigms in **Section 6 (Conclusion and Future Direction)**.
>
> ### 2. Response to W2: Non-Optimal Visualizations (Task vs. Model Capability)
>
> We fully agree with the reviewer that instruction or data characteristics can sometimes lead to layout issues. However, we implemented specific measures during dataset construction and evaluation to mitigate this:
>
> 1.  **Feasibility Check:** We required annotators to write valid reference code for every task. Tasks where a human expert could not produce an optimal layout were removed from the benchmark.
> 2.  **Evaluation Criteria:** Our evaluation prompt explicitly defines low scores for layout issues to ensure fair penalization. For example:
>     * *Score 0:* Significant overlap between subplots, titles, axis labels, or other elements that severely affects readability.
>
> #### **Case Study: Error Case 3 (Figure 32)**
> To address your specific concern, we analyzed Error Case 3 (Figure 32) in the Appendix, which suffered from overlapping x-axis labels and legends. The instruction required a composite visualization of spending amounts and card types across cities.
>
> * **Feasibility:** We have added the human-written reference code for this specific task as **Figure 33** in the updated Appendix. It demonstrates that a clean layout is achievable.
> * **Data Reasoning:** In the provided dataset, the top 8 cities account for over 70% of the transaction volume. Selecting these top cities ensures sufficient information density without overcrowding the x-axis.
> * **Model Limitation:** Crucially, the instruction did not mandate displaying *all* cities. We provided a detailed data description in the prompt. We believe a capable model should use this information to make flexible design choices (such as filtering key data) to avoid overlaps. Therefore, the overcrowding in Figure 32 reflects a lack of reasoning capability in the model, rather than a limitation of the task itself.
>
> ### 3. Response to W3 & Q2: LLM-as-a-Judge Quality
>
> We acknowledge the potential bias when relying on LLM-based judges. To address this, we designed our evaluation framework to be rigorous and objective:
>
> 1.  **Structured Scoring:** To minimize the influence of model preference, we adopted a fine-grained scoring system. We defined specific criteria for distinct aspects using fixed scores (0/1 or 0/1/2). The final score is obtained by summing these aspect scores (Compliance Metrics: max 4; Quality Metrics: max 10). This additive approach reduces the subjectivity often associated with holistic scoring.
> 2.  **Reliability Analysis:** We discuss judge reliability extensively in **Section 5.3** and **Appendix H**. Specifically, **Tables 4, 8, and 9** demonstrate the correlation between human annotators and various models (Gemini-2.5-Pro, Claude-4-Sonnet, and ChatGPT-4o).
> 3.  **Judge Selection:** Our experiments indicated that at the time of this study, only **Gemini-2.5-Pro** demonstrated sufficient capability to evaluate these complex tasks, showing high consistency with human judgments across all aspects. Other models lacked the necessary precision for this benchmark.
> 4.  **Robustness:** To further ensure stability, the reported results are the average of 5 independent evaluation runs by Gemini-2.5-Pro.
>
>
>
> **Gemini-2.5-Pro Cohen's Kappa Agreement Scores** (Table 4)
>
> | Compliance Metrics | | | | Quality Metrics | | | | |
> | :---: | :---: | :---: | :---: | :---: | :---: | :---: | :---: | :---: |
> | **Layout** | **Type** | **Visual** | **Task** | **Clarity** | **Layout** | **Color** | **Text** | **Format** |
> | 0.90 | 0.78 | 0.72 | 0.80 | 0.73 | 0.71 | 0.69 | 0.65 | 0.73 |

---

### Official Review · Reviewer_8EMQ · 2025-11-01

**Soundness:** 2
**Presentation:** 2
**Contribution:** 2
**Rating:** 4
**Confidence:** 5

**Summary:**

The paper introduces PLOTCRAFT, a benchmark for LLM-based generation of complex visualization code, featuring single-turn generation and multi-turn refinement tasks across three difficulty tiers and a rich taxonomy of intents and chart types. The authors also release SYNTHVIS-30K, a sizable dataset sourced from real-world data, and fine-tune PLOTCRAFTOR, which achieves competitive performance with a relatively small model size. The evaluation covers 24 models with detailed analyses and discussion. Results highlight current deficiencies of large language models on visualization code generation and demonstrate the effectiveness of SYNTHVIS-30K for complex plot generation.

**Strengths:**

- Relevant topic: The paper addresses an emerging area in LLM-based complex visualization generation and outlines a research gap that is worth exploring. It also proposes a benchmark to facilitate further study in this space.

- Use of real-world data: The SYNTHVIS-30K dataset is reasonably large and built from real data sources, which helps enhance its practical value.

- Evaluation design: The paper presents experiments across 24 models and provides an analysis of their behavior in visualization code generation. The comparison between single-turn and multi-turn settings offers some insights into refinement effects. The reported alignment between the automatic evaluator and human judgments provides an initial indication of its reliability.

**Weaknesses:**

- **Undefined notion of “multi-chart” generation:** The paper does not clearly define what constitutes *multi-chart* or *multi-plot* generation. If it merely refers to placing several plots side-by-side, the novelty and significance are limited. A more meaningful definition should clarify whether the goal involves cross-chart coherence, shared data semantics, or insight-driven multi-view coordination. It is also unclear whether the dataset considers the relationship between generated charts and the underlying data insights.

- **Evaluation fairness concern:** PlotCraftor is fine-tuned on its own benchmark and then evaluated on the same benchmark. This setting favors the proposed model and limits fair comparison with other models that were not adapted to the dataset.

- **Mismatch between claims and actual content on layout:** Although the Introduction frames layout design as a key challenge, the paper provides limited technical treatment of layout generation. The notion of *layout* seems reduced to basic Matplotlib subplot configuration, but a more comprehensive explanation of what layout entails (e.g., spatial organization, visual hierarchy, annotation distribution, cross-chart alignment) is missing.

- **Inaccurate dataset comparison (Table 1):** The paper states that prior datasets do not include multi-chart scenarios, but ChartArXiv does contain multi-chart examples. The comparison table should be corrected to reflect this.

- **Reliability of the multi-turn evaluation setup:** The multi-turn refinement scenario is valuable and aligns with how humans iteratively improve charts. However, the evaluation relies on a single model judge. Given that chart quality is inherently subjective, relying on one judge may bias the results. A more robust approach could involve a *multi-judge ensemble* or voting across different models to reduce bias and increase scoring reliability.

- **Limited exploration of edit-based refinement:** The idea of refining existing chart code is interesting, but current refinement requires regenerating the entire code. A more intelligent and realistic approach would support *edit-based* modifications—e.g., adjusting key parameters, fixing specific components, or applying localized patches—rather than full code rewriting.

- **Dataset construction details insufficiently described:** Since the dataset is positioned as a main contribution, the construction methodology needs greater transparency. The paper lacks detail on dataset design criteria, annotation guidelines, quality control processes, and whether multiple annotators were involved to ensure correctness and consistency.

**Questions:**

1. **Clarification on multi-chart definition:**
   How do the authors formally define *multi-chart* generation in this work? Does it involve cross-chart coordination or insight-driven composition, or is it limited to placing multiple plots within a single figure?

2. **Evaluation fairness:**
   Since PlotCraftor is fine-tuned on the proposed benchmark and then evaluated on it, how do the authors ensure fair comparison with other models that were not trained or adapted to this dataset? Would zero-shot and fine-tuned baselines be considered for a fairer evaluation?

3. **Reliability of single-judge scoring:**
   The multi-turn refinement setting is interesting, but chart quality can be subjective. Why did the authors rely on a single model judge? Have the authors considered using multiple judges or an ensemble voting mechanism to improve scoring robustness?

4. **Dataset construction transparency:**
   As the dataset is a primary contribution, could the authors provide more details on the construction criteria, annotator workflow, and quality control procedures? Was multi-annotator verification performed to ensure correctness and consistency of refinement instructions?

---

> ### Author Response · Authors · 2025-11-21
>
> We sincerely thank Reviewer 8EMQ for the valuable and constructive feedback. We appreciate the time you took for such an in-depth analysis. We offer the following clarifications in response to your concerns.
>
>
>
> ### 1. On the Definition of "Multi-Chart" (W1, Q1)
>
> We thank the reviewer for this important question. We would like to clarify that our definition of "multi-chart" is **not** merely "placing several plots side-by-side."
>
> We define "multi-chart" complexity across our Medium and Hard task categories in **Section 3.2 (Task Complexity)**. We also provide a detailed breakdown in **Appendix A.3** and show clear examples in **Figure 13**.
>
> Our definition relies on two key dimensions:
> 1.  **Composite Chart:** A single figure integrating multiple plot types (e.g., a bar chart with a line plot overlay).
> 2.  **Multi-Panel Grid:** A figure with multiple subplots (e.g., 2x2, 2x3).
>
> As defined in Appendix A.3, these dimensions determine the complexity:
>
> > • **Medium:** Tasks involve creating a composite visualization. This can be either: (a) a single figure that integrates multiple plot types or variables [...] or (b) a multi-panel grid [...] composed of simple, individual plots.
>
> > • **Hard:** Tasks demand the creation of a complex, multi-panel grid [...] wherein each individual subplot is itself a composite chart.
>
> In summary, Medium tasks require one of these dimensions, while Hard tasks require both simultaneously.
>
> Furthermore, these tasks require high-level visualization goals and cross-chart coherence, not just simple stacking. For example, the Hard task in Figure 13 and Figure 14 requires a 3x3 grid where each subplot analyzes the temporal evolution (1990-2020) of CO2 emissions and environmental metrics for different countries, all aimed at identifying trends and statistical insights.
>
> ### 2. On the Evaluation Fairness Concern (W2, Q2)
>
> We understand your concern regarding evaluation fairness and the separation of datasets. We would like to clarify that PlotCraftor was NOT trained on the PlotCraft benchmark.
>
> Our training dataset, `SynthVis-30K`, was created using a multi-agent framework. During this synthesis, we ensured **no data overlap** with the held-out `PlotCraft` benchmark. We detail this process and our data leakage prevention measures in **Section 4.1**. Therefore, the evaluation ensures a fair comparison, as all models were tested on unseen data.
>
> To further demonstrate PlotCraftor's generalization ability, we also conducted experiments on two other standard benchmarks: **VisEval** and **PandasPlotBench**. As shown in **Table 3**, PlotCraftor achieves strong performance, proving its capability beyond our specific dataset. For a clear comparison, the performance of the base model (Qwen3-Coder-30B-A3B) is also reported in both Table 2 and Table 3.
>
>
> ### 3. On the Reliability of Single-Judge Scoring (W5, Q3)
>
> This is an excellent and constructive suggestion. We share your concern and, during our research, we did experiment with multiple model judges and a multi-judge ensemble.
>
> We discuss the reliability of different judge models in **Section 5.3** and **Appendix H**. Specifically, **Table 4, Table 8, and Table 9** show the correlation between human judges and various models (Gemini-2.5-Pro, Claude-4-Sonnet, and ChatGPT-4o).
>
> Our experimental results showed that at the time this work was completed, only **Gemini-2.5-Pro** was sufficiently capable of judging these complex tasks effectively. The other models lacked the required capability. Consequently, a multi-judge ensemble did not produce reliable results, as the overall performance was degraded by the weaker models.
>
> Gemini-2.5-Pro demonstrated a high agreement with human judges. To further enhance the robustness of our findings, all scores reported in **Table 2** are the average of 5 separate evaluation runs using Gemini-2.5-Pro as the judge.
>
> **Gemini-2.5-Pro Cohen's Kappa Agreement Scores** (Table 4)
>
> | Compliance Metrics | | | | Quality Metrics | | | | |
> | :---: | :---: | :---: | :---: | :---: | :---: | :---: | :---: | :---: |
> | **Layout** | **Type** | **Visual** | **Task** | **Clarity** | **Layout** | **Color** | **Text** | **Format** |
> | 0.90 | 0.78 | 0.72 | 0.80 | 0.73 | 0.71 | 0.69 | 0.65 | 0.73 |

---

> ### Author Response · Authors · 2025-11-21
>
> ### 4. On Dataset Construction Transparency (W7, Q4)
>
> Thank you for this suggestion. The `construction criteria` and `annotator workflow` are already detailed in the existing **Appendix B**, and we have added **Appendix B.5** to specifically elaborate on our `quality control procedures`.
>
> * **Construction (Appendix B):** We outline a rigorous two-phase process where three visualization experts generated and filtered tasks from a candidate pool of nearly 3,000 entries. The final set covers 7 visualization intents and over 50 chart types, ensuring instructions are abstract and goal-oriented.
> * **Quality Control (New Appendix B.5):** To guarantee robustness, we implemented a triple-verification strategy. Three independent annotators validated every multi-turn task for Error Existence (visual confirmation) and Instruction Consistency. Only tasks achieving unanimous consensus were included in the final benchmark.
> ### 5. On the Mismatch Regarding Layout (W3)
>
> We agree with the reviewer that "layout" is a complex term. We would like to clarify that we do not reduce layout generation to only "basic Matplotlib subplot configuration."
>
> We detail our full evaluation criteria and the specific prompt used for judging in **Appendix C**. As shown there, our assessment of layout quality is comprehensive and considers multiple factors, including element sizing, spacing, overall visual balance, and annotation overlapping.
>
> ### 6. On the Inaccurate Dataset Comparison (Table 1) (W4)
>
> Thank you for pointing out this error in our comparison. We have corrected Table 1 in the revised paper.
>
> The corrected entry for ChartArXiv now reflects this accurately:
>
> | Benchmarks | \# of Test Instances | Composite Types | Multiple Subplots | Multi-Turn | Evaluation Metric |
> | :--- | :--- | :--- | :--- | :--- | :--- |
> | CharArXiv | 93K | ✗ | ✓ | ✗ | MLLM Score |
>
> ### 7. On the Limited Exploration of Edit-Based Refinement (W6)
>
> We fully agree with the reviewer that `edit-based modifications` are a valuable research direction, and we plan to explore this approach in our future work.
>
> In this study, we focused on full code regeneration because it aligns with the current prevalent workflow of general-purpose LLMs (e.g., web-based chat interfaces). In these scenarios, visualization code is typically concise and self-contained. For most users, copying the complete updated script is often more convenient and less error-prone than manually applying patches or diffs. Therefore, we believe our current setup effectively captures a highly practical and common user experience.
>
> Moving forward, we plan to extend PlotCraft to include explicit edit-based tasks, requiring models to generate code diffs or localized parameter adjustments. This will be an essential next step for supporting advanced, fine-grained code editing agents.

---

### Comment · Area_Chair_vF4t · 2025-11-22

Dear Authors and Reviewers,

I would like to thank the authors for providing detailed rebuttal messages on time.

To reviewers: I would like to encourage you to carefully read all other reviews and the author responses and engage in an open exchange with the authors. Please post your first response as soon as possible within the discussion time window. Ideally, all reviewers will respond to the authors, so that the authors know their rebuttal has been read.

Best regards,
AC

---

### Author Response · Authors · 2025-12-02
**Rebuttal summary**

Dear Area Chair and Reviewers,

We sincerely thank all reviewers for their thoughtful and constructive feedback. The reviewers unanimously recognized the importance of our work, highlighting our contributions as a **"timely and significant benchmark"** and praising the high quality of the manuscript's presentation and methodology.

We have provided comprehensive responses to all questions raised by the four reviewers. Although we have not received follow-up responses, we believe we have thoroughly addressed every concern. The reviewers' feedback primarily focused on enhancing the paper's transparency, robustness, and clarity, all of which we have incorporated into the revised manuscript.

Based on the reviewers' comments, we have made several major improvements to our paper:

##### New Experiments & Analysis
* We conducted a detailed inter-rater reliability analysis for our human evaluation, reporting Fleiss' Kappa scores and 95% Confidence Intervals. The results demonstrate "Almost Perfect" agreement among human raters, confirming the high quality of our ground-truth data. (mentioned by Reviewer cAYw)
* To enhance the robustness of our findings, we have added variance estimates (mean ± std) across multiple evaluation runs for our main results in a new **Table 10**. (mentioned by Reviewer cAYw)
* We incorporated new case studies in the Appendix, including a human-generated reference chart (**Figure 33**) to prove task feasibility and an example of MLLM judge failure (**Figure 34**) to transparently show its limitations. (mentioned by Reviewer BuCn, gbQN)

##### Clarity & Transparency Enhancements
* We have provided full transparency on our dataset construction and synthesis pipeline, adding details on quality control procedures (**Appendix B.5**), agent models, and associated costs. (mentioned by Reviewer 8EMQ, gbQN)
* We added a prominent discussion on the limitations of automated evaluation, incorporating suggested literature to provide a more balanced perspective. (mentioned by Reviewer gbQN)
* We clarified key definitions, the scope of our benchmark (detailed instructions vs. abstract goals), and data licensing to improve the paper's precision. (mentioned by Reviewer 8EMQ, BuCn, cAYw)


Sincerely,

The Authors

---

### Note · Authors · 2026-01-30

I have read and agree with the venue's withdrawal policy on behalf of myself and my co-authors.

---

### Meta-Review · Area_Chair_bz7r · 2026-01-04

**Summary:**

The submission introduces PlotCraft, a benchmark targeting complex visualization code generation, together with the SynthVis-30K dataset and a fine-tuned model, PlotCraftor. Reviewers broadly agree that the benchmark addresses a timely problem, is substantially more challenging than prior work, and is supported by extensive empirical evaluation across a wide range of models.

The main points of discussion focus on evaluation reliability (in particular, the reliance on a single LLM judge), clarity of task definitions (especially the notion of multi-chart generation and layout), dataset construction transparency, synthesis cost and reproducibility, and fairness in comparing a fine-tuned model against untuned baselines. The authors' responses are generally detailed and technically sound, and they address most reviewer concerns. However, several issues remain as acknowledged limitations rather than being fully resolved, notably the dependence on a single proprietary judge, the restricted backend scope, and the cost and reproducibility of the synthesis pipeline.

The initial reviewer scores range from one score of 4 to three scores of 6. No reviewers have responded to the rebuttal.

The most substantive negative concern lies in the definition and significance of "multi-chart" generation. While the authors clarify their definition in the rebuttal and point to Appendix A.3 and Figures 13 - 14 in the original submission, Reviewer 8EMQ's concern persists that the proposed "multi-chart" tasks may largely reduce to placing multiple plots within a single figure. After check with the original paper, I believe that this interpretation holds, the novelty and conceptual significance of the benchmark are weakened, as the increased difficulty may primarily stem from a higher number of multi-subplot instances rather than from fundamentally new requirements such as cross-chart semantic coherence or insight-driven coordination. Under this view, the benchmark risks being perceived as an incremental extension of existing datasets (e.g., VisEval) rather than a qualitatively new evaluation setting.

Another major concern raised by multiple reviewers is the heavy reliance on a single automated judge (Gemini-2.5-Pro). Although the authors provide evidence of strong alignment with human evaluations and justify their choice by showing weaker performance of alternative judges, this setup still raises concerns about bias, stability, and sensitivity to judge-specific failure modes. The reliance on a proprietary model further complicates reproducibility. While the authors discuss cost-reduced alternatives (e.g., Gemini-2.5-Flash) in the context of data synthesis, the effectiveness of these alternatives for evaluation or training is not empirically validated in the current work.

Overall, the paper is well written and addresses an important and timely problem in LLM-based visualization generation. However, the contribution appears largely incremental relative to existing benchmarks, with increased task difficulty driven mainly by more complex compositions rather than a clearly distinct problem formulation. In addition, the reliability and reproducibility of the evaluation framework would benefit from further strengthening before the benchmark can serve as a long-term, community-wide standard.

I do not recommend acceptance of this paper at this time.

**Reviewer Concerns:**

Reviewer concerns addressed by the rebuttal:
* Definition of multi-chart tasks (Reviewer 8EMQ)
* Evaluation fairness regarding model training (Reviewer 8EMQ)
* Dataset construction transparency and quality control (Reviewer 8EMQ, cAYw)
* LLM-as-judge justification and reliability analysis (Reviewers 8EMQ, BuCn, gbQN, cAYw)
* Clarification of instruction granularity and task feasibility (Reviewer BuCn)
* Corrections and minor issues (Reviewers 8EMQ, cAYw)


Reviewer concerns that are still outstanding:
* Conceptual novelty of "multi-chart" generation (Reviewer 8EMQ):
Despite clearer definitions, it remains unclear whether the benchmark introduces fundamentally new challenges beyond increased compositional complexity (e.g., multi-subplot layouts). The concern that "multi-chart" tasks may largely reduce to placing multiple plots in a single figure—and thus represent an incremental extension of prior work—was not fully resolved.
* Dependence on a single proprietary judge (Reviewers 8EMQ, BuCn, gbQN, cAYw):
Although justified empirically, the evaluation still relies on a single closed-source model. Risks related to bias, long-term stability, and reproducibility remain, and no fully validated alternative evaluation strategy is provided.
* Reproducibility and cost of the synthesis pipeline (Reviewers gbQN, cAYw):
The rebuttal disclosed cost and model usage but did not fundamentally mitigate the barrier posed by reliance on expensive proprietary models, particularly for reproducing or extending the dataset.
* Limited backend scope (Reviewer cAYw):
The benchmark remains restricted to Python or Matplotlib, with no empirical evidence that findings generalize to other visualization languages or ecosystems.
* Edit-based refinement and realism of workflows (Reviewer 8EMQ):
The authors acknowledged edit-based refinement as future work, but the limitation remains unaddressed in the current contribution.

**Reviewer Scores:**

Based on the rebuttal and the nature of the remaining concerns, I do not expect any reviewer to have materially changed their original scores had they participated fully in the discussion.

---

### Decision · Program_Chairs · 2026-01-26

Reject